# Enhanced Antibacterial Property of Facet-Engineered TiO_2_ Nanosheet in Presence and Absence of Ultraviolet Irradiation

**DOI:** 10.3390/ma13010078

**Published:** 2019-12-22

**Authors:** Kenichiro Hayashi, Kosuke Nozaki, Zhenquan Tan, Kazuhisa Fujita, Reina Nemoto, Kimihiro Yamashita, Hiroyuki Miura, Keiji Itaka, Satoshi Ohara

**Affiliations:** 1Graduate School of Medical and Dental Sciences, Tokyo Medical and Dental University, Bunkyo-ku, Tokyo 113-8549, Japan; khysfpro@tmd.ac.jp (K.H.); k.fujita.fpro@gmail.com (K.F.); r.nemoto.fpro@tmd.ac.jp (R.N.); yama-k.bcr@tmd.ac.jp (K.Y.); h.miura.fpro@tmd.ac.jp (H.M.); 2State Key Laboratory of Fine Chemicals, School of Petroleum and Chemical Engineering, Dalian University of Technology, Panjin 124221, China; 3Institute of Biomaterials and Bioengineering, Tokyo Medical and Dental University, Chiyoda-ku, Tokyo 101-0062, Japan; 4Joining and Welding Research Institute, Osaka University, 11-1 Mihogaoka, Ibaraki, Osaka 567-0047, Japan; ohara@jwri.osaka-u.ac.jp

**Keywords:** TiO_2_, facet engineering, photocatalytic activity, antibacterial activity

## Abstract

Titania (TiO_2_) has attracted much attention recently for reducing bacterial diseases by the generation of reactive oxygen species (ROS) under UV irradiation. However, demand for higher photocatalytic activity due to higher recombination of electron and hole remains. The aims of this study were to make titania with higher antibacterial property and show the mechanisms of the bactericidal effect. In this study, we hydrothermally synthesized TiO_2_ nanosheets (NS) with highly-oriented structures. Samples were divided into five groups, depending on the fluorine/titanium ratio in the raw material, namely NS1.0, NS1.2, NS1.5, NS1.8, and NS2.0. Facet ratio and nanosheet size increased with an increase of fluorine/titanium ratio. The photocatalytic activity of TiO_2_ nanosheet was assessed by the generation of ROS. Hydroxyl radicals and superoxides were generated efficiently by ultraviolet light irradiation on NS1.5 and NS1.0, respectively. Antibacterial activity against *Streptococcus mutans* was assessed in the presence and absence of UV irradiation; NS1.0 showed superior antibacterial properties compared to commercially available TiO_2_ nanoparticles, under both conditions, due to the oxidation of intracellular components and cell membrane. These results together suggested TiO_2_ nanosheet induced bacterial cell death by oxidation, and TiO_2_ facet engineering resulted in enhancement of both photocatalytic and antibacterial activities of TiO_2_.

## 1. Introduction

Bacterial infections constitute a major cause of chronic infections and mortality. Although antibiotics have been proposed as the preferred treatment tool for bacterial infections, considering the emergence of multidrug-resistant bacterial strains, much attention has been recently focused on titania (TiO_2_) materials, having antibacterial activities that are irrelevant in most of the antibiotic resistance mechanisms [1]. TiO_2_ has been well documented for its self-cleaning action over the past years, and its photocatalytic activity has also been majorly discussed [2].

In the field of dentistry, powdery TiO_2_ is widely used for disinfecting removable dentures and bleaching teeth. Its antibacterial activity has been reported to be induced by reactive oxygen species (ROS), generated on the TiO_2_ surface by UV irradiation [3,4]. ROS, including hydroxyl radicals and superoxides, attack the bacterial cell membranes and destroy them, thereby deactivating the bacterial cell [5]. TiO_2_ nanoparticle has also been reported to penetrate the cell membrane and generate ROS in the cytoplasm, hence resulting in cell death.

Recently, a variety of modifications in the photocatalysts has been reported to enhance their efficiency against bacteria, considering some well-identified limitations of titania-based photocatalysts, such as higher recombination of electron and hole and limited solar light absorption [5]. Silver-modified TiO_2_ has been reported to delay the recombination time and show high antimicrobial property [6,7]. Also, facet-engineered surface and interface design are some of the recent advances for enhancing photocatalytic materials [8]. Titania, synthesized by the traditional method, has mainly {101} facet [9], whereas Yang et al. had succeeded in synthesizing titania with a large number of exposed {001} facets due to suppression of the crystal growth of {001} plane by adding hydrogen fluoride (HF) during the synthesis of titania [10]. Since there are various limitations in the use of hydrogen fluoride, and risk of residues remain, when used as a biomaterial [11], TiO_2_ crystals have been synthesized by alternative HF-free synthetic strategies as well [12,13,14,15,16]. We reported a novel method to synthesize titania nanosheet without using hydrogen fluoride, along with its enhanced degradation of organic dyes [17]. Density functional theory (DFT) calculations showed the {101} and {001} surfaces to form heterojunction that would enhance the photocatalytic activity by spatial separation of electron and hole [18]. Since superoxide and hydroxyl radicals, generated in aqueous suspension, are linearly correlated with electrons and holes [19], an appropriate relative ratio of {001} to {101} facets is bound to play a pivotal role in the antibacterial property. Liu et al. had compared ROS generation with antibacterial property, against *Escherichia coli* and *Staphylococcus aureus*, using TiO_2_ with {101}/{001} ratio of 49, 11.5, 1.78, and 0.087. Although they demonstrated TiO_2_ with a {101}/{001} ratio of 1.78 ({001} (facet percentage was 36%) to be the best, showing the highest ROS and most potent antibacterial performance [20], due to the lack of facet ratio of over 36%, the optimal facet ratio is still unknown. Furthermore, the bactericidal effect of TiO_2_ nanosheet against *Streptococcus mutans*, which is the pathogenic bacteria for dental caries and denture plaque, is still unknown since the photocatalytic activity varies depending on the type of pollutants oxidized/reduced by the facet-engineered TiO_2_ [18,21,22,23].

In this study, a series of highly-tailored TiO_2_ nanosheets, with various {001}/{101} facet ratios, were prepared by hydrothermal method, and the percentages of {001}/{101} facet ratio and size were tuned by a delicate control of fluorine/titanium ratio. To achieve an optimal facet ratio for bactericidal activity against *Streptococcus mutans*, we evaluated the generation of ROS and the survival rate of bacterial cells. The underlying mechanism was further investigated to correlate with oxidative stress. As a result, TiO_2_ nanosheets with a 52% {001}/{101} facet ratio showed superior bactericidal effects compared to other nanosheets and commercially available nanoparticles. The bactericidal effect was induced not only in the presence of UV irradiation but also in the absence. Cytotoxic mechanisms have been suggested to involve oxidation of cell membranes and intracellular components followed by cell membrane disruption. This study was expected to provide new insights for the development of effective antibacterial biomaterials based on facet engineering.

## 2. Results

### 2.1. Characterization of TiO_2_ Nanosheet

The X-ray diffraction (XRD) diffractogram of precipitates is shown in Figure 1a. All diffraction peaks were in good agreement with the standard XRD profile of anatase-type TiO_2_ (PDF# 21-1272). Representative transmittance electron microscope (TEM) images of TiO_2_ nanosheet (NS) are shown in Figure 1b. Samples were divided into five groups, depending on the fluorine/titanium ratio (1.0, 1.2, 1.5, 1.8, and 2.0) in the raw material, namely NS1.0, NS1.2, NS1.5, NS1.8, and NS2.0. NS1.0 was the smallest, both in thickness and length, and its size increased with the increase in the F/Ti ratio (Figure 1c). The {001} facet ratio of NS1.0 was the smallest and increased with an increase in the F/Ti ratio (Figure 1d).

### 2.2. Effect of TiO_2_ NS on UV-Stimulated ROS Generation

Disodium terephthalic acid (NaTA) and nitro blue tetrazolium (NBT) assay detected changes in hydroxyl radical and superoxide generation in the TiO_2_ mixture after 60-min exposure of different sizes of nanosheet to UV light with intensity 2.5 mW/cm^2^ at 365 nm (Figure 2a,b). With the increase in the F/Ti ratio, hydroxyl radical generation first increased and then decreased, as shown in Figure 2a. NS1.5 showed superior hydroxyl radical generation compared to other NS and nanoparticle (NP). NP used in this study were commercially available anatase-type TiO_2_ nanoparticles with average size 7 nm, which was synthesized without a capping agent. Due to the difficulty in analyzing facet ratio by morphology, we estimated the facet ratio of 6% from the other report [24]. However, superoxide generation first decreased and then increased, with the increase in the F/Ti ratio (Figure 2b). NP showed superior superoxide generation compared to NS groups.

### 2.3. Evaluation of Antibacterial Properties

Results of the antibacterial activity of NS against *S. mutans* under UV irradiation clearly showed effective growth inhibition compared to that with NP (Figure 3a). Survival rates highly depended on the concentrations and crystal size. NS 1.0 exhibited excellent antibacterial properties even at low concentrations (0.05 and 0.1 mg/mL) compared to other NS and NP. Because the positive relationship between the ROS generation and antibacterial property of TiO_2_ NS was not observed, NS1.0, NS2.0, and NP were further studied in detail.

In order to evaluate the UV effect on antibacterial property, we investigated the cell viabilities with NS1.0 in the presence and absence of UV irradiation. The irradiated group was found to exhibit superior antibacterial action compared to the non-irradiated group (Figure 3b). Next, we evaluated the survival rate of bacteria in the presence of L-histidine, which has been shown to scavenge a ROS in many studies. L-histidine itself did not affect cell viability since no inhibition or growth was found in dark conditions (Figure 3b). Although UV irradiation decreased the viable cells in the presence or absence of TiO_2_, histidine rescued the viability of bacteria.

### 2.4. Evaluation of Oxidation State of Bacteria

After 30-min exposure to TiO_2_ NS and NP, in the presence of UV irradiation, oxidation was observed in the intracellular components as per the fluorescence images of 2′,7′-dichlorodihydrofluorescein diacetate (H2DCFDA) (Figure 4a). H2DCFDA is a fluorescent dye after removing the acetate groups by intracellular esterase and oxidation. While TiO_2_ NS1.0 and NS2.0 induced the oxidation of intracellular components even under dark conditions, TiO_2_ NP did not (Figure 4b). The control groups without TiO_2_ did not show oxidation in either the presence or absence of UV irradiation conditions. Figure 5 shows the C-11 4,4-difluoro-5-(4-phenyl-1,3-butadienyl)-4-bora-3a,4a-diaza-s-indacene-3-undecanoic acid (BODIPY) fluorescence images and oxidation of bacterial cell membranes after 30-min exposure to TiO_2_ NS and NP in the presence (Figure 4a) and absence of UV irradiation conditions (Figure 4b). C-11 BODIPY is a fluorescent fatty acid analog taken up by the cell membrane and used for the measurements of lipid oxidation in the living cell. TiO_2_ NS1.0 induced the oxidation of bacterial cell membranes in both the presence and absence of UV irradiation conditions. However, NS2.0 oxidized only in the presence of UV irradiation condition, whereas NP did not cause cell membrane oxidation at all.

### 2.5. Evaluation of Cell Membrane Damage

Cell membrane damage was assessed using SYTO9 (live-cell) and propidium iodide (PI, dead cell), as shown in Figure 6. TiO_2_ induced cell membrane damage, regardless of being in the presence or absence of UV irradiation. For quantification, the red and green areas were measured using Image J software, respectively, and the results are shown in Figure 7. TiO_2_ caused cell membrane destruction in both the presence and absence of UV irradiation conditions. NS1.0 and NS2.0 showed severe damage compared to NP.

## 3. Discussion

To control the facet ratio of titania, we used ammonium hexafluorotitanate as the fluorine source to prevent the residues of HF after synthesis. TEM images clearly showed the suppression of {001} facet growth and promotion of crystal growth along a- and b-axis direction, with an increase in the F/Ti ratio. As a result, the dose of capping agents against TiO_2_ played a pivotal role in the regulation of {001}/{101} facet ratio. These results indicated the HF-free strategies were acceptable to synthesize the highly-oriented TiO_2_ nanosheet.

The photocatalytic activity was assessed by the generation of hydroxyl radicals and superoxide. The total amount of superoxide and hydroxyl radical showed TiO_2_ NS1.0 to be superior compared to the other nanosheet specimens, although it was inferior compared to TiO_2_ NP. Quantification of ROS is important for understanding photodegradation mechanisms since ROS are primary intermediates of photocatalytic reactions in aqueous solutions under aerobic conditions [19]. Generation of hydroxyl radical and superoxide is reported to be largely dependent on the {101} and {001} facets of TiO_2_ nanosheet with heterojunctions [18]. TiO_2_ nanosheet, synthesized in this study, also indicated a facet ratio-dependent generation of ROS.

The antibacterial property of TiO_2_ NS1.0 was the highest compared to that of other NS and NP and was possibly caused by the oxidation of bacterial cell membranes and DNA. Although mechanisms of toxicity of the nanomaterials are yet to be understood, bactericidal mechanisms of metal oxides have been proposed to include: (1) ROS generation, (2) metal ion release, (3) nanomaterial accumulation on the membrane surface, and (4) internalization of nanomaterials [25]. Among them, the major toxicity of metal oxide nanomaterials is attributed to ROS-induced damage owing to a linear regression between the bactericidal effect and average concentration of total ROS [26].

We also confirmed the ROS-induced bactericidal effect of TiO_2_ NS by radical scavenger assay using L-histidine. L-histidine has been recognized as a scavenger of hydroxyl radical and singlet oxygen [27] and has been reported to decrease the rate of disinfection in TiO_2_ nanoparticles [28]. Antibactericidal effect of TiO_2_ NS1.0 and 2.0 under UV irradiation was suppressed by supplementation with L-histidine, and the cell viability was recovered to that in the control group. L-histidine also suppressed the anti bactericidal effect under dark conditions. Since TiO_2_ in the absence of UV irradiation conditions cannot generate ROS in aqueous phase, but can cause oxidation of C-H bond by electron transfer to an acceptor site [21], we estimated the bacterial cell wall to be oxidized by electron transfer, and then bacterial cell death was induced.

We further showed the oxidation of cell membrane and intracellular components by TiO_2_ NS and the subsequent destruction of the cell membranes. TiO_2_ NP induced oxidation of the cell membrane and intracellular components only in the presence of UV irradiation. However, TiO_2_ NS induced oxidation in both the presence and absence of UV irradiation conditions. Live/dead staining also indicated that the cell membrane damage by TiO_2_ NP was induced only in the presence of UV irradiation, whereas TiO_2_ NS induced damage in both the presence and absence of UV irradiation conditions. The cytotoxicity mechanisms in the presence and absence of UV irradiation conditions have been reported to involve ROS generation and nanoparticle attachment to the cell membrane, respectively [29]. Since TiO_2_ can cause oxidation not only by ROS but also by electron transfer, bacterial cell membranes are thought to be directly oxidized by TiO_2_ NS. In this experiment, the electron transfer mechanism in the dark of the NS was unknown, but it would be possible to develop better antibacterial biomaterials by elucidating the mechanism in the future.

## 4. Materials and Methods

### 4.1. Synthesis of TiO_2_ Nanosheet

TiO_2_ nanosheets were synthesized, as previously described [17]. Briefly, ammonium hexafluorotitanate and titanium butoxide (Sigma-Aldrich Co., St Louis, MO, USA, 97%) were used as starting agents. Ammonium hexafluorotitanate was dissolved in 5 M HCl and then mixed with titanium butoxide. To control the {001}/{101} facet ratio, titanium/fluorine molar ratio in the mixture was set as 1.0 (NS1.0), 1.2 (NS1.2), 1.5 (NS1.5), 1.8 (NS1.8), and 2.0 (NS2.0). The solution was placed in a high-pressure reaction vessel and allowed to react at 180 °C for 6 h. After hydrothermal synthesis, precipitates were washed thrice with distilled water, once with methanol, and eventually dispersed in distilled water.

### 4.2. Characterization of TiO_2_ Nanosheet

Crystal structure, morphology, and chemical components of precipitates were characterized using an X-ray diffractometer (XRD, D8 advance, Bruker AXS GmbH, Karlsruhe, Germany) and transmittance electron microscope (TEM, H-7100, Hitachi High-Technologies Corporation, Tokyo, Japan). The length and thickness of the nanosheet were measured from TEM images (n = 3). The exposed {001} facet ratio (*P*_001_) was calculated as follows:(1)$$S_{001}=2\left(L-\frac{d}{\tan\theta }\right)$$
(2)$$S_{101}=\frac{2d}{\sin\theta }\left(2L-\frac{d}{\tan\theta }\right)$$
(3)$$P_{001}=\frac{S_{001}}{S_{001}+S_{101}}$$
where *L* and *d* are the average length and thickness of nanosheets, respectively, from TEM images; *S*_001_ and *S*_101_ are the areas of all {001} and {101} facets, respectively, in a TiO_2_ single crystal; *P*_001_ is the exposed percentage of {001} faces; *θ* = 68.3° is the theoretical value of the angle between (001) and (101) facets of anatase [30].

### 4.3. Measurement of Reactive Oxygen Species

#### 4.3.1. Measurement of Hydroxyl Radical

Fluorescence of 2-hydroxy terephthalic (2-HTA) acid has been reported to reflect the oxidation of disodium terephthalic acid (NaTA) by hydroxyl radicals [31]. TiO_2_ nanosheets (NS1.0, NS1.2, NS1.5, NS1.8, and NS2.0), dispersed in 20 mM NaTA (Tokyo Chemical Industry Co. Ltd., Tokyo, Japan) solution, were poured into 24-well dishes and irradiated for 60 min at intensity 2.5 mW/cm^2^ with UV rays from a source with wavelength 365 nm (FPL27BLB, Sankyo Electric Co., Ltd., Tokyo, Japan). In this study, TiO_2_ nanoparticles were used as a control group (NP, STS-01, Ishihara Sangyo Kaisya, Ltd., Osaka, Japan). After UV irradiation, the solution was centrifuged, supernatants were collected, and fluorescence intensity (Ex 340 /Em 460 nm) was measured using a fluorescence microplate reader (Wallac Arvo MX, Perkin Elmer Co., Ltd., Waltham, MA, USA).

To prepare the calibration curves for hydroxyl radicals, 2-HTA (Tokyo Chemical Industry Co. Ltd., Tokyo, Japan) was dissolved in distilled water and diluted up to representative concentrations. The absorbance of the solution was measured using a fluorescence microplate reader. The corresponding hydroxyl radical concentration was estimated by the following reaction, assuming that the trapping efficiency of terephthalic acid (TA) for hydroxyl radicals is 80% [32].NaTA + ^•^OH → 2-HTA

#### 4.3.2. Measurement of Superoxide

Formazan has been reported to reflect the oxidation of nitro blue tetrazolium (NBT) by superoxide [33]. TiO_2_ nanosheets and nanoparticles, dispersed in 1 mM NBT (Tokyo Chemical Industry Co. Ltd., Tokyo, Japan) solution, were poured into 24-well dishes and irradiated for 60 min at intensity 2.5 mW/cm^2^ with UV at a wavelength of 365 nm. After UV irradiation, the solution was centrifuged (8000× *g*), and the supernatant was discarded. Precipitates were dissolved in dimethyl sulfoxide (DMSO, FUJIFILM Wako Pure Chemical Corporation, Osaka, Japan), and absorbance of the solution at 580 nm was measured using a microplate reader (Model680, Bio-Rad Laboratories Inc, Hercules, Berkeley, CA, USA).

To prepare calibration curves for superoxide, pure formazan from photochemical reactions was obtained according to a previous report [34]. Briefly, a reaction mixture containing 27 μM riboflavin, 17 mM methionine (FUJIFILM Wako Pure Chemical Corporation, Osaka, Japan), and 1 mg/mL NBT in 50 mM potassium phosphate buffer (pH 7.8) was incubated for 1 h at room temperature. The purple precipitates were washed five times with distilled water and ethanol and dried at 60 °C overnight to determine the weight. A standard solution was prepared by dissolving pure formazan in DMSO and diluting up to the representative concentrations; eventually, absorbance at 580 nm was measured using a microplate reader.

### 4.4. Evaluation of Antibacterial Activity

#### 4.4.1. Evaluation of Antibacterial Properties of TiO_2_ Nanosheet

Antibacterial properties of TiO_2_ nanosheet were evaluated using *Streptococcus mutans* (MT8148), which was kindly provided by Prof. Ichiro NAKAGAWA of Tokyo Medical and Dental University, and the antibacterial test using these bacteria have been previously reported [35,36]. The details of these bacteria are described in other articles [37,38]. *S. mutans* has been reported as one of the popular bacterial species in denture plaque [39]. *S. mutans* was cultured in brain-heart infusion (Bacto Brain Heart Infusion, Becton, Dickson and Company, city, MD, USA) for 24 h at 37 °C in an ambient atmosphere. Bacterial suspensions were centrifuged at 1700× *g* for 5 min, supernatants were discarded, and saline was poured to obtain an optical density of 0.5 using a turbidity meter (Biochrom WPA CO8000 Cell density meter, Cambridge, UK). The bacterial suspensions in saline were poured into 24-well dishes, mixed with TiO_2_ nanosheets (NS1.0, NS1.2, NS1.5, NS1.8, and NS2.0), and irradiated with 365 nm light (2.5 mW/cm^2^) for 30 min. After UV irradiation, the viable cells were evaluated by the ATP measurement method (Promega BacTiter-Glo™ Microbial Cell Viability Assay Kit, Promega Corporation, Madison, WI, USA), and the luminescence was measured using a luminometer (GloMax Navigator system, Promega Corporation, Madison, WI, USA). The survival rate of bacteria was calculated as follows:(4)$$\mathrm{Signal}\;\mathrm{to}\;\mathrm{noise}\;\mathrm{ratio}\;\left(S:N\right)=\frac{Mean\;signal-mean\;of\;background}{Standard\;deviation\;of\;background}$$
(5)$$\mathrm{Survival}\;\mathrm{rate}=\frac{S:N\;of\;experimental\;group}{S:N\;of\;samples\;without\;TiO_{2}\;before\;irradiation}A\times 100$$

#### 4.4.2. Antioxidation Assay

To evaluate the antibacterial effect of ROS, the antibacterial assay was performed in the presence of L-histidine, which has been reported as a radical scavenger [33]. After the pre-culture of *S. mutans*, the cell suspension was mixed with NS1.0, 2.0, or NP, and then irradiated with UV light under the same conditions as in antibacterial assay, in the presence of 20 mM L-histidine. After UV irradiation, viable cells were evaluated by the adenosine triphosphate (ATP) measurement method.

#### 4.4.3. Oxidative Stress on *S. mutans* by TiO_2_ Nanosheet

In order to confirm the oxidative stress of TiO_2_ nanosheet on *S. mutans*, we evaluated the oxidation state of the cell membrane and DNA of bacteria using NS1.0, NS2.0, and NP. After the pre-culture of *S. mutans*, its cell membrane and DNA were stained using 4,4-difluoro-5-(4-phenyl-1,3-butadienyl)-4-bora-3a,4a-diaza-s-indacene-3-undecanoic acid (BODIPY 581/591 C11, Life technologies-Invitrogen, Carlsbad, CA, USA) and H2DCFDA (Life technologies-Invitrogen) at 37 °C for 30 min, respectively. After staining, the cell suspension was centrifuged at 1700× *g* for 5 min, the supernatant was removed, and the cells were re-suspended in brain heart infusion (BHI) solution at 37 °C for 30 min, to render the dye responsive to oxidation. After replacing BHI with saline, the cell suspension with TiO_2_ (0.4 mg/mL) was irradiated with UV under the same conditions of the antibacterial experiment. After irradiation, the cells were centrifuged at 1700× *g* for 5 min, and the supernatant was removed. After washing with saline thrice, the cells were observed under a fluorescence microscope (IX71, Olympus Corp., Tokyo, Japan).

#### 4.4.4. Cell Membrane Damage

To confirm cell membrane damage by TiO_2_ nanosheet, we evaluated cell viability by live and dead cell staining methods using NS1.0, NS2.0, and NP. After the pre-culture of *S. mutans*, cell suspension in saline with TiO_2_ was irradiated with UV under the same conditions as in the antibacterial experiment. After irradiation, cell suspensions were stained with propidium iodide (PI) and SYTO9 (LIVE/DEAD™ BacLight™ Bacterial Viability Kit, Life technologies-Invitrogen) and observed under a fluorescence microscope. From fluorescence images, the area ratio of live- and dead-bacteria was measured using imaging software (ImageJ, NIH) and calculated subsequently as follows:(6)$$\mathrm{Live}-\mathrm{to}-\mathrm{dead}\;\mathrm{bacteria}\;\mathrm{ratio}=\frac{Area\;of\;live\;cell}{Area\;of\;live\;cell+Area\;of\;dead\;cell}\times 100$$

### 4.5. Statistical Analysis

Statistical analysis was performed using the Mann–Whitney U test, with Bonferroni corrections for hydroxyl radicals, superoxides, survival rate, and live-to-dead bacteria ratio. Results with p-values below 0.05 were considered statistically significant.

## 5. Conclusions

Bactericidal effect of titania nanosheet, with highly oriented facet against *Streptococcus mutans*, was induced not only in the presence but also in the absence of UV irradiation. Titania nanosheet (NS1.0) with 52% {001}/{101} facet ratio showed superior bactericidal effect compared to the other nanosheets and commercially available nanoparticles. The cytotoxicity mechanisms were suggested to involve the oxidation of cell membrane and intracellular components, followed by cell membrane destruction. This reactivity might contribute to developing antibacterial materials using titania, even in the absence of UV irradiation conditions.

## Figures and Tables

**Figure 1 materials-13-00078-f001:**
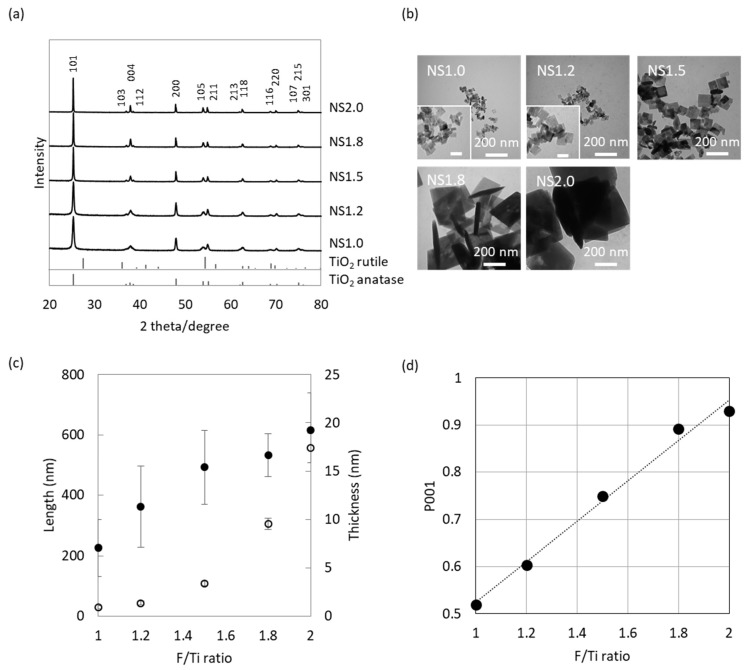
Characterization of TiO_2_ (titania) nanosheet. (**a**) XRD patterns of TiO_2_ NS1.0, 1.2, 1.5, 1.8, 2.0, and nanoparticle (NP). (**b**) TEM images of TiO_2_ NS (nanosheet). The inset is a high magnification of samples. Scale bar: 50 µm (**c**) Length and thickness of each nanosheet measured from TEM images. Open circle: thickness (nm), Closed circle: Length (nm). (**d**) The ratio of {001} facet (P_001_) calculated from length and thickness.

**Figure 2 materials-13-00078-f002:**
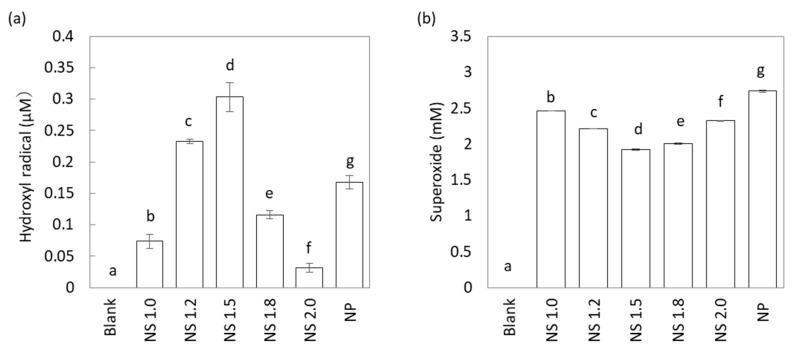
The amount of hydroxyl radical (**a**) and superoxide (**b**) generated by UV irradiation. Different letters show significant differences (*p* < 0.05).

**Figure 3 materials-13-00078-f003:**
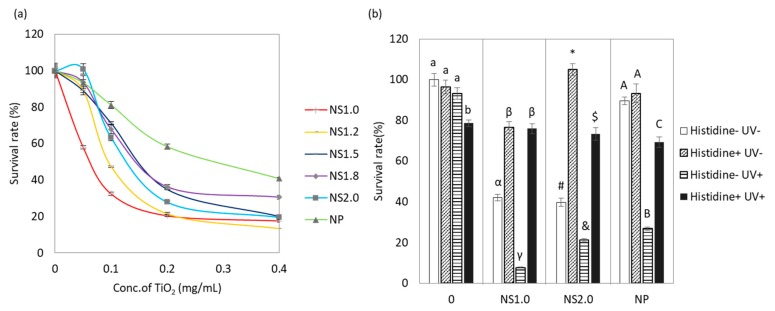
Antibacterial property of TiO_2_ nanosheet. (**a**) The survival rate of *S. mutans* cultured with TiO_2_ NS and NP under UV irradiation. (**b**) Antioxidant effect on the survival rate of *S. mutans* cultured with NS1.0, 2.0, and NP in both the presence and absence of UV irradiation conditions. Different letters in the same material groups show significant differences (*p* < 0.05).

**Figure 4 materials-13-00078-f004:**
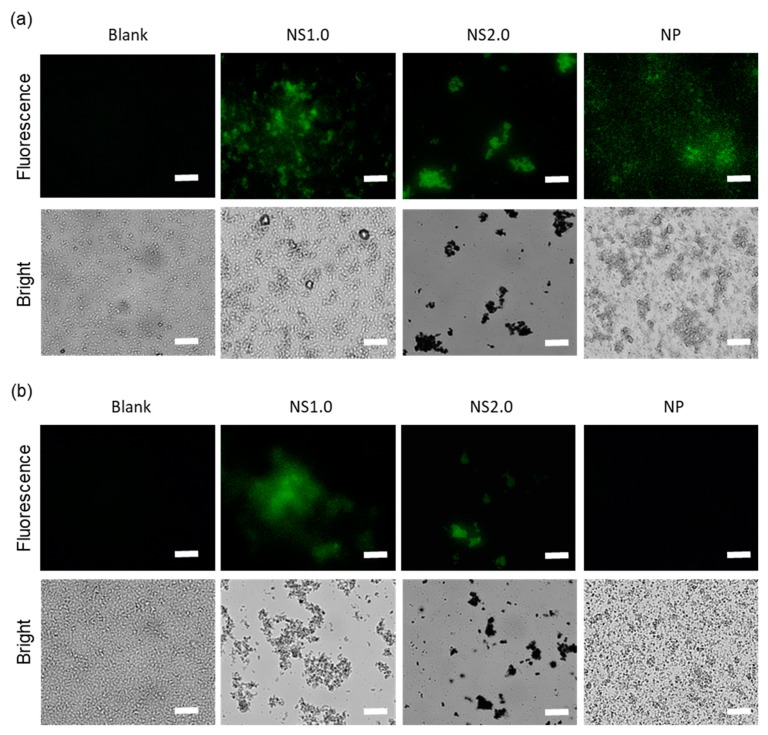
Intercellular ROS (reactive oxygen species) accumulation in *S. mutans* loaded with H2DCFDA (2′,7′-dichlorodihydrofluorescein diacetate) in the presence (**a**) and absence (**b**) of UV irradiation conditions. Cells were observed under a fluorescence microscope (Upper panel) and a phase-contrast microscope (Lower panel). Scale bar: 250 μm.

**Figure 5 materials-13-00078-f005:**
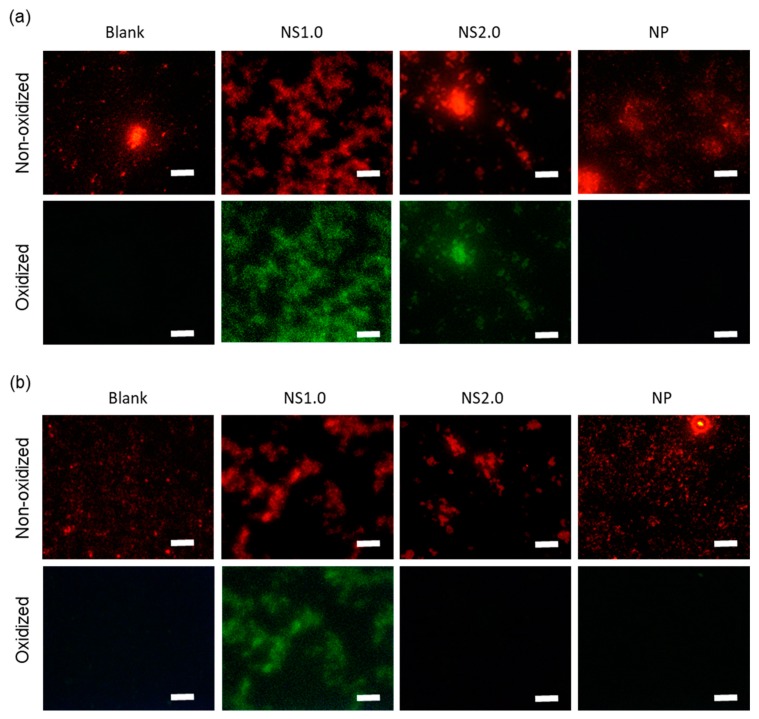
Lipid peroxidation of *S. mutans* loaded with C11-BODIPY581/591 in the presence (**a**) and absence (**b**) of UV irradiation conditions. Upper panel: non-oxidized image. Lower panel: oxidized image. Scale bar: 250 μm.

**Figure 6 materials-13-00078-f006:**
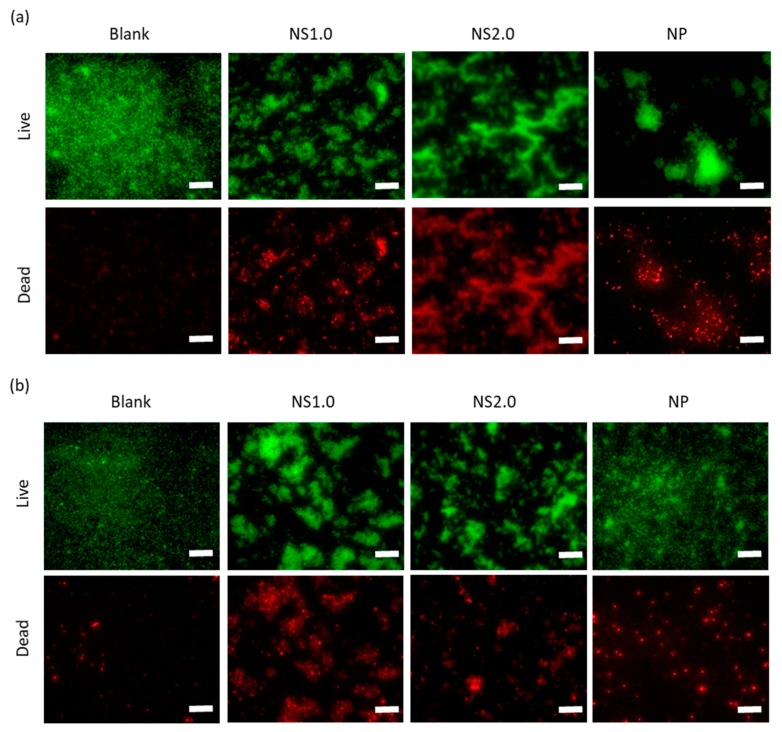
Live/dead staining images of *S. mutans* under the presence (**a**) and absence (**b**) of UV irradiation conditions. Green fluorescence shows live cells stained with SYTO9 (upper panel), while red fluorescence shows dead cells stained with propidium iodide (PI) (lower panel). Scale bar: 250 μm.

**Figure 7 materials-13-00078-f007:**
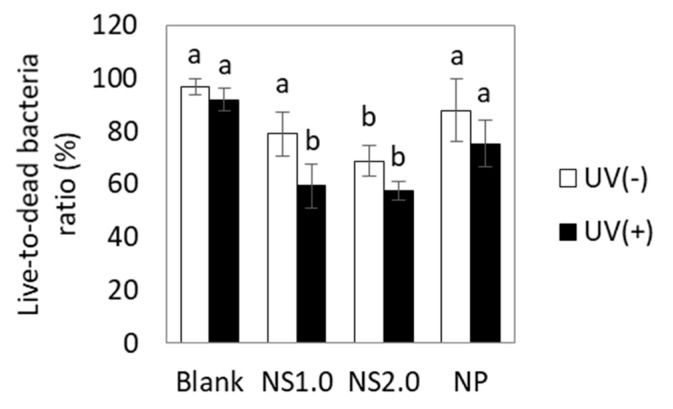
Live-to-dead bacteria ratio. Different letters in the same material groups show significant differences (*p* < 0.05).

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
