# Peer review of "Enhanced Antibacterial Property of Facet-Engineered TiO2 Nanosheet in Presence and Absence of Ultraviolet Irradiation"

_materials, 2019, doi:10.3390/ma13010078_

Round 1

Reviewer 1 Report

In this manuscript, the authors reported hydrothermally synthesis of five groups of TiO2 nano-sheets with different fluorine/titanium ratio. Photocatalytic activity of TiO2 nano-sheet was then evaluated by the generation of ROS. Antibacterial activity of TiO2 nano-sheet against Streptococcus mutans was assessed in presence and absence of UV irradiation. Both Photocatalytic and antibacterial activities were compared with a commercially available TiO2 nanoparticles.

The concept of this work is great, however the following minor points need to be revised before this work can be published:

Minor Comments:

In the result section, page 3 line 98 under “2. Effect of TiO2 NS on UV-stimulated ROS generation”, it is recommended that a short specification the UV light to be added. e.g. UV intensity 2.5 mW/cm2 at 365 nm. Figure 1d, “Ratio of {001} facet calculated from length and thickness.” The calculation details should be added to the experimental section. In figure 2 and 3, [page 3, line 101; anatase-type TiO2 nanoparticles], the specification details of commercial NP e.g. size, F/Ti ratio should be added and compared with the synthesized NS. Figure 3b, those a, b, c and d letters on top of the columns charts seems not to be correct. If a is no histidine and no UV, b with histidine and no UV, c no histidine and UV and d both histidine and UV, then how the control sample has a,a,a,b and NP a,a, b,c rather than NS1.0 and NS2.0 with a,b,c,d. When histidine was introduced for the first time in page 4 line 117, the role of histidine as radical scavenger (mentioned in discussion) should be briefly mentioned. Correct “2′`,7′-dichlorodihydrofluorescein” to “2′,7′-dichlorodihydrofluorescein”. The same when 2′,7′-dichlorodihydrofluorescein diacetate (H2DCFDA) was introduced in the page 5 line 128,129 the role and properties should be briefly mentioned e.g. H2DCFDA is a fluorogenic probe under oxygen-reactive condition and the same for the BODIPY fluorescence how the color of oxidized and non-oxidized are changing should be mentioned.

Reviewer 2 Report

The present article reports an experimental investigation on the photocatalytic and antibacterial activity of TiO2 nanosheets. The nanosheets were prepared by hydrothermal synthesis by tuning the fluoride/titanium ratio in the precursors to control the {001}/{101} facet ratio and, consequently, the functional properties of the materials. Standard characterization techniques were employed, such as X-ray diffraction, transmission electron microscopy, and energy dispersive X-ray spectroscopy. The photocatalytic activity was assessed by the generation of reactive oxygen species (ROS) under UV light irradiation, namely hydroxyl and superoxide radicals, which showed two different activity trends based on the {001}/{101} facet ratio. The antibacterial activity of two selected TiO2 nanosheets samples was tested against Streptococcus mutans both under UV irradiation and in the dark and compared to commercial TiO2 nanoparticles. The authors reported a bactericidal effect of the nanosheets in both conditions, on the contrary to TiO2 nanoparticles (only active upon UV irradiation), which was interpreted in terms of oxidation of C−H bonds in the cell membranes.

TiO2 nanomaterials have been extensively studied in various applications thanks to their photocatalytic activity. Their use to remove bacterial infections is thus well-justified and agrees with the cited literature. The manuscript is well-organized and clear (only few minor English mistakes are present). The characterization techniques employed to investigate the TiO2 nanosheets are adequate. Moreover, the ROS generation tests give useful insights in order to investigate the antibacterial activity of the TiO2 nanosheets. The manuscript is thus of potential interest for the readers of Materials. Few points should be revised before publication, as described below.  

Figure 1(a) shows a decrease of the FWHM of the diffraction peaks with the increase of the F/Ti ratio. The authors may calculate the crystallite size by the Scherrer formula. In addition, the TEM images in Figure 1(b) should possibly be reported with the same magnification to allow a better comparison of the morphology of the different samples. Figure 2(a) clearly shows that the sample NS1.5 generated the highest amount of hydroxyl radicals, thus higher than TiO2 nanoparticles as well as than the NS1.0 and NS2.0 samples. On the other hand, this sample has not been considered for the antibacterial activity tests against mutans in the following sections (Figures 4−7). The same experiments may be performed also for the NS1.5 sample in order to better discuss the ROS bactericidal effect. The initial part of the discussion section (p. 8, lines 161−182) may be summarized because it repeats some concepts already mentioned in the introduction and conclusion sections. In line 87, the authors should briefly mention the meaning of the sample names; similarly, in line 97, the same should be done to explain the meaning of “NaTA” and “NBT”. Few minor writing mistakes should be amended, for example: line 19, “lower recombination rate of electrons and holes” instead of “higher recombination of electron and hole”; line 49, “limited solar light absorption” instead of “lower solar sensitivity”; line 56, “we reported” instead of “we had reported”; line 67 “Streptococcus mutans” in italic; line 83, “XRD diffractogram” instead of “XRD spectrum”; line 112, “excellent” instead of “particularly excellent”; line 135, “only in the presence of” instead of “in only presence of”; line 211, insert the reference number for Tan et al.; line 215, “set as” instead of “decided as”. Some papers mention potential toxic effect of TiO2 nanoparticles to eukaryotic cells. It could limit the potential antibacterial using of this material. Are there any information about toxicity of these samples to human cells?

Reviewer 4 Report

This manuscript describes comparatively the enhanced antibacterial property of facet-engineered TiO2 nanosheet in presence and absence of ultraviolet irradiation. Moreover, the study aims to obtain titania with higher antibacterial property and show the mechanisms of the bactericidal effect. TiO2 was hydrothermally synthesized under the form of nanosheets with highly oriented structures. Antibacterial activity against Streptococcus mutans was assessed in presence and absence of UV irradiation. The article is well documented and the explanations enough clear.

The author used relevant characterization methods to highlight the surfaces important properties, considering the final envisaged applications.

There are some corrections/ observations regarding the manuscript:

1a): On the graphic, the standard XRD profile of anatase-type TiO2 was inversed with that of rutile profile. Please inverse their name. Also, it will be very helpful to add the facet orientation above the peaks (at least for <101> and <001>) It’s already well known that the UV irradiation change drastically the surface energy for TiO2 (through the modification of contact angle). Considering this aspect, how can authors explain that the bactericidal effect of titania nanosheet is exhibiting not only in presence, but also in the absence of UV irradiation.

  In my opinion, this manuscript is suitable to be published in “Materials” after minor revision.

Reviewer 5 Report

Author must revise manuscript according to following comments:

Author must show total ROS generation using Kits or dyes for quantitative data with statistics (such as H2DCFDA, DAFFMDA etc) Author must show statistics in biological data. For statistics use *,**, *** symbols to denote T-Test or annova significance values. author must show figure 4 and 5 quantitative data.

Round 2

Reviewer 5 Report

Author have not revised manuscript according to my previous comment.

I request author again to revise manuscript according to my previous comment given below.

Author must show total ROS generation using Kits or dyes for quantitative data with statistics (such as H2DCFDA, DAFFMDA etc) Author must show statistics in biological data. For statistics use *,**, *** symbols to denote T-Test or annova significance values.